behaviour/cognition

dogs, word learning, object names, long-term memory

**Author for correspondence:**
Shany Dror
e-mail: shanymd@gmail.com

# Acquisition and long-term memory of object names in a sample of Gifted Word Learner dogs

Shany Dror[1], Ádám Miklósi[1,2], Andrea Sommese[1], Andrea Temesi[1] and Claudia Fugazza[1]

[1]Department of Ethology, Eötvös Loránd University, Budapest, Hungary
[2]MTA-ELTE Comparative Ethology Research Group, Budapest, Hungary

SD, 0000-0003-4039-6217

Dogs with a vocabulary of object names are rare and are considered uniquely gifted. In a few cases, these Gifted Word Learner (GWL) dogs have presented cognitive skills that are functionally similar to those of human infants. However, the acquisition rate of new object names and the ability of GWL dogs to form long-term memories of those is unknown. In this study, we examine the ability of six GWL dogs to acquire the names of new objects in a short period and to retain those in their long-term memory without post-acquisition exposures. In Experiments 1 and 2, the dogs were tested on their ability to learn, during social interactions with their owners, the names of 6 and 12 new toys respectively, in one week. In Experiments 3 and 4, the dogs' memory of these objects was tested after one and two months. GWL dogs typically learned the names of the new objects and remembered those. We suggest that dogs with knowledge of object names could be a powerful model for studying mental mechanisms related to word acquisition in a non-human species.

## 1. Introduction

During the vocabulary spurt, defined as a sudden increase in the rate of word learning, infants shift from learning to produce one or two new words a week, to up to nine new words in a day [1]. One of the proposed explanations of this quantitative shift is a qualitative change in the involved learning mechanisms. Instead of learning specific associations between sounds and objects, infants develop an understanding that words represent entities in their environment [2].

To establish functional word learning, it is crucial to retain the name–object links beyond the initial exposure. It is often

problematic to compare studies examining the ability of infants to form long-term memory of words, as those measure different retention intervals of either production or comprehension, different exposure protocols, infants of different ages, and the use of nouns or verbs. Woodward *et al.* [3] found that both 13-month-old infants (before the vocabulary spurt) and 18-month-old infants (that have started the vocabulary spurt) learned a new object name and retained its memory for 24 h. Childers & Tomasello [4] described how 2.5-year-old infants learned a novel noun or verb and retained the memory of the newly learned word for one week. Markson & Bloom [5] reported that 3- and 4-year-olds retained the memory of a newly learned noun for up to one month. Apes have been traditionally used as a model for studying language-related processes ([6], for review) and some studies have been carried out on other species too, including marine mammals [7,8] and parrots [9]. However, unlike for these species, bred or kept in captivity, the human niche is dogs' natural environment [10]. Therefore, dogs are recognized as a representative model for comparative studies on the evolution of social-cognitive systems [10]. In addition, family dogs are often raised in an environment that is very similar to that of infants, and dog owners were documented speaking to their dogs in a manner that shares acoustical [11] as well as grammatical and syntactical [12] similarities with infant-directed speech.

Only a few studies have documented the ability of dogs to learn object names [13–15] and recent findings suggest that this ability is present in only a few uniquely gifted dogs (Gifted Word Learner (GWL) dogs) [16]. In two independent studies, each conducted with a single GWL dog [14,17], the dogs demonstrated an understanding that one word can represent different objects, which can serve as a proof for categorical comprehension. One of these studies [14] also described how the dog was able to learn the names of over a thousand objects by being exposed to a very intensive and prolonged training protocol, including 5–6 h of professional training each day, for 3 years. Recently, a study involving two GWL dogs found that, during a brief playful interaction with their owners, the dogs were able to link new names to new objects after only four exposures [18]. The dogs managed to maintain the memory of these new links for at least 2 min, but it declined rapidly after delays of 10 min. Beyond this, little is known about the rate with which GWL dogs acquire new object names so that these are retained over longer periods.

To examine the acquisition of object names, we tested the ability of six dogs with previous vocabulary knowledge of objects to learn the name of 6 and 12 new objects in one week (Experiments 1 and 2, respectively). To test memory consolidation, in Experiments 3 and 4, we tested these dogs' memory of the new object names after one and two months.

## 2. Methods

Ethical permission for conducting this study was obtained from The Institutional Committee of Eötvös Loránd University (N. PE/EA/691-5/2019). All owners gave informed consent to participate in the study with their dogs.

### 2.1. Subjects

For 2 years, we searched for dogs of any age and breed (or mix breed) that knew the name of their toys through media announcements and word of mouth during international seminars, conferences and lectures given by the authors. By these means, we recruited $N = 6$ border collies, living as family dogs (3 females, 3 males, average age = 3.6 years, s.d. = ±2.7). These dogs have all proven their knowledge of names of objects in a previous study [16] aimed to evaluate their existing vocabulary and their ability to acquire new object names over a prolonged period (three months). When we started the current study, all the dogs knew over 26 names of objects. They proved this knowledge by successfully retrieving each toy upon the owner's request when the toy was placed in a different room, out of the owners' view, among other named toys [16].

### 2.2. Experimental set-up

As the dogs that participated in this study live in different countries, the experiments were conducted using an online video streaming software (StreamYard®). To control for the clever Hans effect [13], the owners were sitting in one room while the toys were placed in another room, out of view from them. A camera positioned in each room broadcasted the experiment so that the experimenters could always see both the owner and the dog in real-time. This test was part of the Genius Dog Challenge (www.geniusdogchallenge.com) social media campaign, aimed at locating more dogs with knowledge of object names. Thus, parts of the tests were broadcasted live over the Genius Dog Challenge YouTube channel.

## 2.3. Learning procedures

### 2.3.1. Experiment 1

Each dog owner received six new dog toys and s/he had one week to teach his/her dog their names. The names of the new toys were chosen randomly from suggestions of people following the project over social media. The only constraint when choosing the names of the toys was that they could not sound similar to any of the dogs' existing toys.

The owners were allowed to freely interact with their dogs and teach them the new object names as they saw fit. For all owners, this included short sessions in which they were playfully interacting with their dogs. In each play session, the owners selected one of the new toys, presented it to the dog, and, while repeating its name, encouraged the dog to bite the toy, tossed it in the air, and asked the dog to fetch it from a pile of other objects or retrieve it from another room. A video demonstrating such a play session can be found at: https://www.youtube.com/watch?v=G5esin56yGo. Four owners reported spending 0.5 h or less each day playing and naming the toys (owners of Max, Rico, Nalani and Squall). One owner reported playing the game for 1.5 h on average per day (owner of Whisky), and one reported playing the game for approximately 2.5 h (owner of Gaia). The time the owners dedicated to playing with their dogs, with the named toys, was dependent on their personal availability. Play sessions often occurred as part of the normal interaction between dog and owner, as a dog-initiated play with a toy, and not as formal training. Therefore, the amount of time the owners dedicated to playing also varied according to the time they spent at home (which was also affected by local lockdowns practised in different countries to prevent the spread of the coronavirus COVID-19).

The success of the dogs in learning the name of these toys was tested on the 7th day (see below).

### 2.3.2. Experiment 2

This experiment was carried out two or three weeks after Experiment 1 was completed, in a similar manner to Experiment 1, but this time the owners were requested to teach their dogs the names of 12 new toys in one week. The success of the dogs in learning the name of these toys was tested on the 7th day. During this week, five of the owners reported spending approximately the same time playing with their dogs, as stated for Experiment 1. Only one dog owner (of the dog Gaia) increased the time spent playing with her dog to approximately 5.5 h per day. After the test on the 7th day, the dogs had an additional week to play with the toys before they were stored away until they were reintroduced in the memory tests of Experiments 3 and 4 (see below).

## 2.4. Testing procedures

### 2.4.1. Experiment 1 test

The six new toys were scattered on the floor with 10 old toys that were randomly chosen from the dog's collection of named toys (overall 16 toys on the floor). This was done to decrease the possibility that the dog will retrieve the correct toy by chance. The area in which the toys were scattered was approximately 2 m in diameter. In the test trials, the owner sat in a different room and asked the dog to retrieve each new toy by pronouncing its name. The toys were requested in a randomized order with each new toy tested two times (overall 12 trials). When only three new toys remained on the floor, the owner placed the toys the dog had retrieved back, so the overall number of toys from which the dog chose always varied from 16 to 14 (out of which between six and four were new toys). Parts of Experiment 1 and Experiment 2 (see below) were broadcasted and are available on YouTube. The last three trials appearing in these broadcasts were not used for purposes of data analysis, as they did not meet our criteria of keeping the chance level at 25%. These three trials were properly repeated after the broadcast ended.

### 2.4.2. Experiment 2 test

The set-up and procedure of this test were similar to those described for Experiment 1. However, in this test, in addition to the 12 new toys that were placed on the floor, there were eight old toys (overall 20 toys). Here again, whenever there were only three new toys left on the floor the owner placed all the toys that the dog retrieved back, always maintaining the overall number of toys from which the dog could choose between 12 to 20 (out of which between 12 and 4 were new toys). Each new toy was tested two times in a randomized order (overall 24 trials).

### 2.4.3. Experiment 3

One-month memory test: out of the 12 toys the dogs were exposed to during Experiment 2, six toys were randomly chosen to be used in this test. One month after the toys had been stored, the dogs' memory of their name was tested. Before the beginning of the test, the owner placed the toys on the floor and allowed the dog to freely interact with those for 3 min while ignoring his/her behaviour. This was done to decrease the possibility that the dog would be over-excited by the presence of the reintroduced toys during the test. After 3 min elapsed, the test was conducted using the same procedure described for Experiment 1, but with eight additional old toys on the floor (overall 14 toys) and requesting each of the reintroduced toys three times (overall, 18 trials). If the dog retrieved an incorrect toy, the experiment was interrupted for a few minutes and the owner removed the requested toy, before continuing to the next trial. This was done as, in previous observations, dogs tended to try to correct their mistake in follow-up trials, by retrieving the previously requested toy, even though the owner had requested a new one, thus causing a chain reaction of mistakes.

Whenever there were only three reintroduced toys left on the floor, the owner would place all the toys back, maintaining the overall number of toys from which the dog could choose between 14 and 12 (out of which between six and four were the reintroduced toys).

### 2.4.4. Experiment 4

Two months memory test: The six remaining toys from Experiment 2 were used for this test. This test was conducted two months after the toys were stored, using the same procedure described for Experiment 3.

### 2.4.5. Testing protocol

In all experiments, the test was carried out as follows:

1. The owner was sitting with the dog in a room.
2. The experimenter told the owner which toy should be retrieved (the order of the toys was randomly determined).
3. The owner asked the dog to retrieve the toy by pronouncing its name (typically: 'Bring < object name>!'.
4. The dog left the room and entered the room with the toys on the floor, where it selected a toy by picking it up and bringing it to the owner.
5. If the dog retrieved the correct toy, the owner praised his/her dog and briefly played with the retrieved toy.
6. If the dog did not retrieve the correct toy:
   a. In Experiments 1 and 2, the trial was repeated, and the repetition was not included in the data analysis. If the dog made a second mistake, the owner retrieved the toy without showing it to the dog or stating its name.
   b. In Experiments 3 and 4, the experiment was interrupted for a few minutes while the owner told the dog to wait in the room and went to remove the toy without showing it to the dog.
7. After the dog (or the owner, in case of repeated mistakes by the dog) retrieved the correct toy, the experimenter instructed the owner to ask for the next toy.

## 2.5. Data collection and analysis

The dogs' correct or incorrect choices of the named toys were coded during the test. Only in one trial in which a dog made an incorrect choice he fetched an old toy. We therefore decided to calculate the chance level in the most conservative way, ignoring the presence of the old toys and always taking into account the lowest number of new toys from which the dog could choose. Consequently, setting the chance level for all experiments at 25%. Binomial tests were used to determine the dog's performance on an individual level.

# 3. Results

In Experiment 1, four dogs successfully retrieved all six new toys ($p \leq 0.001$), and two dogs retrieved five new toys ($p \leq 0.001$) (table 1). As a group, dogs retrieved the correct toy in 84.72% (s.d. ±0.18) of the trials (figure 1). In Experiment 2, two dogs retrieved all 12 new toys ($p \leq 0.001$), and four dogs retrieved 11 new

**Table 1.** Individual and mean results of the dogs in all experiments.

| experiment | dog | number of correct trials out of overall test trials | % | number of new toys retrieved at least once |
|---|---|---|---|---|
| 1 | Max | 12/12 | 100 | 6/6 |
| 1 | Whisky | 12/12 | 100 | 6/6 |
| 1 | Gaia | 12/12 | 100 | 6/6 |
| 1 | Rico | 10/12 | 83.33 | 6/6 |
| 1 | Squall | 8/12 | 66.67 | 5/6 |
| 1 | Nalani | 7/12 | 58.33 | 5/6 |
| 1 | mean | 84.72% (s.d.±0.18) | | |
| 2 | Max | 23/24 | 95.83 | 12/12 |
| 2 | Whisky | 23/24 | 95.83 | 12/12 |
| 2 | Gaia | 21/24 | 87.50 | 11/12 |
| 2 | Rico | 19/24 | 79.17 | 11/12 |
| 2 | Squall | 19/24 | 79.17 | 11/12 |
| 2 | Nalani | 20/24 | 83.33 | 11/12 |
| 2 | mean | 86.81% (s.d.±0.07) | | |
| 3 | Max | 17/18 | 94.44 | 6/6 |
| 3 | Whisky | 13/18 | 72.22 | 6/6 |
| 3 | Gaia | 12/18 | 66.67 | 6/6 |
| 3 | Rico | 13/18 | 72.22 | 6/6 |
| 3 | Squall | 8/18 | 44.44 | 6/6 |
| 3 | Nalani | 3/18 | 16.67 | 3/6 |
| 3 | mean | 61.11% (s.d.±0.27) | | |
| 4 | Max | 12/18 | 66.67 | 6/6 |
| 4 | Whisky | 13/18 | 72.22 | 6/6 |
| 4 | Gaia | 15/18 | 83.33 | 6/6 |
| 4 | Rico | 10/18 | 55.56 | 5/6 |
| 4 | Nalani | 7/18 | 38.89 | 5/6 |
| 4 | Squall | 5/18 | 27.78 | 3/6 |
| 4 | mean | 57.41% (s.d.±0.21) | | |

toys ($p \leq 0.001$). Overall, the dogs retrieved the correct toy in 86.81% (s.d. ±0.07) of the trials (figure 1). In Experiment 3, five dogs successfully retrieved all six toys ($p \leq 0.05$) and one dog retrieved only three toys and did not perform above chance ($p = 0.293$). Overall, the dogs retrieved the correct toys in 61.11% (s.d. ±0.27) of the trials. In Experiment 4, three dogs successfully retrieved all six toys ($p \leq 0.001$), one dog retrieved five toys ($p = 0.03$), and two dogs did not perform above chance (one dog retrieved three toys, $p = 0.5$, and one dog retrieved five toys but overall had 7/18 correct trials, $p = 0.138$) (table 1). As a group, the dogs retrieved the correct toys in 57.41% (s.d. ±0.21) of the trials (figure 1).

# 4. Discussion

Our findings show that the GWL dogs tested here were not only able to learn up to 12 new object names in one week, a learning rate which is comparable to early word acquisition in infants at the beginning of the vocabulary spurt [19] but most of them also maintained a long-term memory of the object names for at least two months. The dogs' learning rate is in line with what was described in a previous study [14]. However, most of the dog owners in our study reported that they spent typically 0.5 h daily playing with the named toys, which is in contrast with the 4–5 h of professional training described in Pilley & Reid [14]. Importantly,

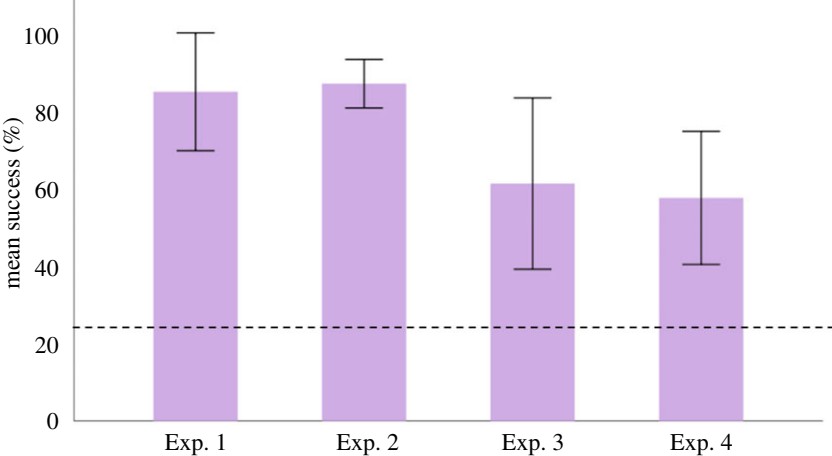

**Figure 1.** The dogs' mean correct percentage (±s.d.) of trials. Experiment 1: learning six toy names in one week (12 trials), Experiment 2: learning 12 toy names in one week (24 trials), Experiment 3: one-month memory test of six toys (18 trials) and Experiment 4: two-month memory test on six toys (18 trials). The chance level for all experiments was set as 25%.

such multiple exposures in an intensive training paradigm prevent us from concluding whether rapid learning of object names took place.

Previous studies on the ability of dogs to learn object names have been restrained by the extremely low sample size of $N = 1$ [13–15] or 2 [18]. Attempts to teach a larger number of typical family dogs the names of objects found that most dogs were not able to do so [16,18,20]. Hence, we decided to include in this study only dogs with an existing vocabulary of object names. Two years before this study, we started searching for such dogs and managed to locate only six subjects living in different countries. Although this sample is a significant increase compared to previous reports, we acknowledge that it is still relatively low. Thus, we presented this experiment as part of a social media campaign (the Genius Dog Challenge) aimed at increasing our sample size in future experiments. Surprisingly, although the campaign received major coverage over popular international media channels, it still resulted in the recruitment of a small group of only about 15 GWL dogs. This supports our previous findings suggesting that the ability to learn object names is rare and present only in a few gifted individuals [16], most of which—but not all—appear to be border collies [13–15,18].

In our previous study [16], the same dogs tested here, all learned between 11 and 37 new toys in a period of three months. We estimated that the rate at which they learned names of new toys was mostly limited by the owners' availability for training. Indeed, in the current study, these dogs all learned between 11 and 12 new toy names in only one week, showing a relatively stable and uniform performance across subjects. We did not attempt to define the limits of the dogs' object word learning abilities, and it is possible that they could learn more than 12 object names in a week. This study also did not limit the owners' interaction, and we, therefore, cannot estimate the minimal amount of exposure that the dogs would have needed to learn. We suggest that future studies should systematically manipulate the learning conditions to estimate the exact amount of exposure dogs need to form and retain long-term memory of object name pairings (e.g. [18]).

Many differences have been pointed out between how infants and GWL dogs comprehend object names. Particularly, it has been emphasized that infants possess a referential understanding of words. This is reflected by the fact that infants do not only associate a new word, like 'sock', with a specific object but rather understand that the word refers to a category of objects, all of which share the same key features [21]. Categorization abilities have been tested in only two GWL dogs [14,17] (one of which is Whisky, also tested in this study). A previous study reported that three GWL dogs comprehended iconic elements of pictures and replicas that referred to corresponding objects. The results of the present study contribute to understanding the ability of GWL dogs to learn object names. By doing so, this study helps pave the way for comparative research shedding light on commonalities and differences between infants' and GWL dogs' understanding of object names. Such research will broaden our view on cognitive aspects related to object names acquisition.

Some theories have been put forward [22] suggesting that human language learning is supported by a combination of domain-general mechanisms, rather than language-domain-specific processes. Domain-general learning mechanisms are likely to be present in a wide range of species, including dogs.

Furthermore, dogs have proved to possess several socio-cognitive traits that are functionally similar to humans' [23,24]. Thus, dogs, with their evolutionary history and development in the human environment provide a useful animal model for studying social cognition (e.g. [25,26]). We suggest that dogs with a vocabulary of object names (GWL dogs) can serve as a comparative model for studying cognitive processes related to object name acquisition.

Ethics. Ethical permission for conducting this study was obtained from The Institutional Committee of Eötvös Loránd University (N. PE/EA/691-5/2019). All owners gave informed consent to participating in the study with their dogs.
Data accessibility. Videos of the test could be found at: https://www.youtube.com/watch?v=53ziA9Q3ESs&list=PLNy4MjoEERsm9cIzG-Nm12kMaR32N6H-q. Table of raw data is added as supplementary material and is also accessible on: Dror, Shany (2021), Dryad, Dataset, https://doi.org/10.5061/dryad.8931zcrr5.
    The data are provided in the electronic supplementary material [27].
Authors' contribution. Conceptualization was done by S.D., Á.M. and C.F.; methodology involved S.D., A.S., A.T and C.F; analysis was done by A.S. and S.D.; writing original draft was done by S.D., Á.M. and C.F.; writing review and editing: all authors reviewed the manuscript.
Competing interests. The authors declare no competing interests.
Funding. This study was supported by the National Brain Research Program (grant no. 2017-1.2.1-NKP-2017-00002). Á.M. received funding from MTA-ELTE Comparative Ethology Research Group (grant no. MTA01 031).
Acknowledgements. We would like to thank Purina® for generously sponsoring this research. We are grateful for the contribution of all the dog owners that participated in this study: Ildiko Gyenes, Isabella Ruzi-Dos Santos Miguel, Sara Infante Diez, Sonja De Laat Spierings, Bobbie Kurvial and Helge O. Svela.

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
