## [Peer Review File · Royal Society Open Science]

Review History

RSOS-210976.R0 (Original submission)

Review form: Reviewer 1

Is the manuscript scientifically sound in its present form?

No

Are the interpretations and conclusions justified by the results?

No

Is the language acceptable?

Yes

Do you have any ethical concerns with this paper?

No

Have you any concerns about statistical analyses in this paper?

Yes

Recommendation?

Major revision is needed (please make suggestions in comments)

Comments to the Author(s)

Dror and colleagues have revised their intriguing paper on the capacities of dogs to learn human labels for objects and to retain these labels for several months. I still have a few comments on the paper as the authors have not quite addressed all my concerns.

L. 19: I still object to the comparison between the dogs and young children. As I noted before, even if children have a very small vocabulary before their 'spurt', the labels are not restricted to one particular object (a "ball" will be used to refer to a number of different balls, and even sometimes the label is over- or under-extended). To argue that dogs are good models for other nonhumans, however, DOES make sense.

L. 34: Here the authors have shifted from early vocabulary acquisition to children well beyond the beginning of the vocabulary spurt...and then shift back to 2 yr olds...the children here are in very different stages and comparisons that are being made are confusing and conflate many issues.

L. 47: Please examine Hirsh-Pasek & Trieman, "Doggerel: Motherese in a new context", paper written ~1982 or so.

L. 53-55: I would again argue that all these dogs learned was a brief association of sound and object; to call this behavior the learning of names is quite a stretch.

L. 59-63: This sentence is quite complex and difficult to parse...please edit.

L. 63: I agree that one dog seems to understand something about categories, which suggests that such learning is possible...but the authors cannot make such claims for the other subjects in the other studies and later for all their subjects.

L. 110: 5.5 hrs/day is intensive training! Which dog was that? Did that dog do better than some of the others? Would be an important bit of information! And why are descriptions of Experiment 2 placed in the paragraph devoted to Experiment 1?

L. 118: How did the experimenters guarantee that the toys were stored away and that the owners did not review things with their dogs? Promises to do so are nice, but people often want their dogs to shine, so....

L. 128: There's a certain amount of 'exclusion' occurring here – dog knows it has fetched X already, so maybe knows to ignore X at least for awhile? Why not put in a mix of fetched and some other old toys, or maybe a brand new, unnamed toy?

L. 163: So here the owner showed the dog the correct toy? Wouldn't that act as a refresher trial?

L. 175ff: To be truly conservative, the chance level should be 1/6 for the first set and 1/12 for the second, assuming that the dog would ignore its old toys and just concentrate on the new ones...

L. 209: In reality, only SOME OF the dogs could perform the task....the authors cannot make a broad generalization.

L. 222ff: Given that the labels are connected to a SINGLE object, rather than a class (e.g., "balls", which is what Pilley and Reid tried to test) for the dogs in THIS study, the authors cannot argue

for anything other than an efficient associative mechanism... That is all that THEY tested. They cannot argue that their dogs would perform as well as others trained and tested in different paradigms. The authors are correct that the POSSIBILITY exists, but their claims for THEIR data are way too strong.

L. 237: Acquiring associations that MAY or MAY NOT be object labels.

As I noted before: Not only are these dogs gifted, but they are all Border collies – dogs specifically BRED to learn quickly and to understand vocal commands given by humans...and even then, not all their subjects do well. Note that collies are no longer bred specifically for herding, but likely for physical standards of “beauty”, so that the genetics no longer hold as strongly, which may be why so few dogs succeed.

I repeat my earlier criticism, that although I DO find the behavior of some of these dogs impressive, I do not agree that their behavior is all that similar to that of children who are learning labels nor that dogs (in general?!--remember only a few Border collies succeed in any kind of label learning at all) can be a model for studying mechanisms involved in human word acquisition. The authors should stick to their truly impressive data (which are impressive!) rather than make additional claims of parallels with human acquisition.

They have backed down a bit and said that more work is needed, but they haven't backed down enough, simply because they do not have the data beyond one dog, Whiskey, to make any such claims – no other dog of theirs has been appropriately tested.

In References: Why are some titles of journal articles capitalized whereas other are not?

Review form: Reviewer 2

Is the manuscript scientifically sound in its present form?

Yes

Are the interpretations and conclusions justified by the results?

Yes

Is the language acceptable?

Yes

Do you have any ethical concerns with this paper?

No

Have you any concerns about statistical analyses in this paper?

No

Recommendation?

Accept with minor revision (please list in comments)

Comments to the Author(s)

In this paper, the authors recruited 6 dogs who had previously demonstrated the ability to associate toys with specific names. They then tested their ability to learn new associations of 6 + 12 objects over a short period (1 week), and whether the dogs could retain those associations 1 and 2 months later without any practice in the meantime. I thought the premise was interesting,

the design was well-executed, and the conclusions were warranted. I did still have a few questions about the specifics (e.g. subject criteria, methodology) which are addressed below in my line-by-line feedback.

Title: I wonder if would be better to say 'acquisition' as opposed to 'learning rate' since your question is broader and categorical (i.e. yes/no did the dogs learn the words in a set amount of time) vs. the specifics that 'learning rate' might call to mind (i.e. how many exposures are needed/learning curves/etc.). Also, rather than 'selected' dogs, maybe 'word-knowledgeable' dogs (or some descriptor that defines how they were selected).

Line 15. Might be useful to clarify here that they are retaining in their memory 'without further practice post-acquisition' or something like that?

Lines 59-64. This sentence is hard to parse and parts are redundant. I think it would be easier to understand if rephrased slightly and broken up into 2 sentences. Maybe something like: "In infants, the ability to rapidly learn new words is thought to mark a shift in their thinking, whereby they now realize that words represent objects [15]. Thus, evidence for comparable rapid learning of object names in dogs, combined with previous evidence showing that dogs with vocabulary knowledge relate labels to categories of items, instead of only associating one word to one object [16], may support the idea that dogs, similar to human children, can learn that words represent items [12]."

Subject criteria - how was their knowledge of over 26 object names verified? E.g. owner report or actually tested by an independent party? Were there nominated dogs who were deemed to not actually meet the criteria? I think it would be helpful to either explain in more detail and/or provide a link to the website (geniusdogchallenge.com), which I found by googling the Genius Dog Challenge social media campaign that you first mention in the experimental setup section directly below. The 'How to Apply' section of the website (assuming that is the criteria the 6 current dogs were held to?) was useful in ascertaining how dogs were deemed to meet the criteria.

Line 107-111: obviously sample size is such that we can't draw any firm conclusions, but would be interesting to know which subjects specifically were trained for an avg of 1.5 h and 2.5-5.5 h, respectively

Lines 144-145: why did the owner remove the requested toy if the dog brought the incorrect one? To make it the same # and ratio of test/control items between dogs? Please explain rationale. And what happened with the incorrectly retrieved toy? Was it replaced, or also removed?

Lines 173-176: what was the rationale for replacing the toys after 3 and 9 trials (as opposed to after every trial)? Why did the # of trials between replacement differ between experiments?

Line 190: It would be helpful to present results of experiment 3 and experiment 4 as you do the first 2 experiments... i.e. when you say XX dogs retrieved all toys, remind the reader of the total number of all toys. So, "In Experiment 3, five dogs successfully retrieved all 6 toys" and "In Experiment 4, three dogs successfully retrieved all 6 toys"

Line 191: how many toys did the dog who didn't perform above chance successfully retrieve?

Line 253: ah great - here is the link to the website! Although this also clarifies that the website was used as a recruitment tool AFTER the recruitment of the 6 dogs in the current study (presumably for future studies)... so I think still useful to add more detail in the methods re: recruitment, at the detail level of the 'how to apply' page of the website

Line 266: consider rephrasing to "The similarities in word learning rate (or acquisition??) and retention capacity between infants and 'word-knowledgeable' dogs further suggests that..."

Decision letter (RSOS-210976.R0)

Dear Ms Dror

On behalf of the Editors, we are pleased to inform you that your Manuscript RSOS-210976 "Learning rate and long-term memory of object names in a sample of selected dogs " has been accepted for publication in Royal Society Open Science subject to minor revision in accordance with the referees' reports. Please find the referees' comments along with any feedback from the Editors below my signature.

Please submit your revised manuscript and required files (see below) no later than 7 days from today's (ie 16-Aug-2021) date. Note: the ScholarOne system will 'lock' if submission of the revision is attempted 7 or more days after the deadline. If you do not think you will be able to meet this deadline please contact the editorial office immediately.

on behalf of Dr Rosalind Arden (Associate Editor) and Kevin Padian (Subject Editor)
openscience@royalsociety.org

Associate Editor Comments to Author (Dr Rosalind Arden):

This is a fascinating piece of work and will be of much interest. The two reviewers would like to see some revisions made and so would we.

The work as dog qua dog is good enough to stand alone without making claims that the dog is a good model for learning about the development of language in children. This is nonsensical - there are a super abundance of children from whom we can gather phenotypic and genetic data that directly bear on human language development - we don't need dogs for that - if you have a particular idea in mind for why the dog is a good model or what we could learn from dogs that we cannot learn from human children and their adult relatives, then please say it.

What is striking and interesting here is the evidence of variation among individuals and of capacity. That this work was done under the restrictions of the pandemic is especially praiseworthy.

I would consider cutting the references to cross-species comparisons, but at the least, please do listen to Reviewer 1 who proposes that the ms still in places goes beyond its evidence. It doesn't need to - the work is great as it is. Reviewer 2 has asked some key questions. Answering those will strengthen the ms.

We hope you will revise this after a careful reading of the Reviewer comments. It will be a splendid paper.

Reviewer comments to Author:

Reviewer: 1

Comments to the Author(s)

Dror and colleagues have revised their intriguing paper on the capacities of dogs to learn human labels for objects and to retain these labels for several months. I still have a few comments on the paper as the authors have not quite addressed all my concerns.

L. 19: I still object to the comparison between the dogs and young children. As I noted before, even if children have a very small vocabulary before their 'spurt', the labels are not restricted to one particular object (a "ball" will be used to refer to a number of different balls, and even sometimes the label is over- or under-extended). To argue that dogs are good models for other nonhumans, however, DOES make sense.

L. 34: Here the authors have shifted from early vocabulary acquisition to children well beyond the beginning of the vocabulary spurt...and then shift back to 2 yr olds...the children here are in very different stages and comparisons that are being made are confusing and conflate many issues.

L. 47: Please examine Hirsh-Pasek & Trieman, "Doggerel: Motherese in a new context", paper written ~1982 or so.

L. 53-55: I would again argue that all these dogs learned was a brief association of sound and object; to call this behavior the learning of names is quite a stretch.

L. 59-63: This sentence is quite complex and difficult to parse...please edit.

L. 63: I agree that one dog seems to understand something about categories, which suggests that such learning is possible...but the authors cannot make such claims for the other subjects in the other studies and later for all their subjects.

L. 110: 5.5 hrs/day is intensive training! Which dog was that? Did that dog do better than some of the others? Would be an important bit of information! And why are descriptions of Experiment 2 placed in the paragraph devoted to Experiment 1?

L. 118: How did the experimenters guarantee that the toys were stored away and that the owners did not review things with their dogs? Promises to do so are nice, but people often want their dogs to shine, so....

L. 128: There's a certain amount of 'exclusion' occurring here – dog knows it has fetched X already, so maybe knows to ignore X at least for awhile? Why not put in a mix of fetched and some other old toys, or maybe a brand new, unnamed toy?

L. 163: So here the owner showed the dog the correct toy? Wouldn't that act as a refresher trial?

L. 175ff: To be truly conservative, the chance level should be 1/6 for the first set and 1/12 for the second, assuming that the dog would ignore its old toys and just concentrate on the new ones...

L. 209: In reality, only SOME OF the dogs could perform the task...the authors cannot make a broad generalization.

L. 222ff: Given that the labels are connected to a SINGLE object, rather than a class (e.g., "balls", which is what Pilley and Reid tried to test) for the dogs in THIS study, the authors cannot argue for anything other than an efficient associative mechanism... That is all that THEY tested. They cannot argue that their dogs would perform as well as others trained and tested in different paradigms. The authors are correct that the POSSIBILITY exists, but their claims for THEIR data are way too strong.

L. 237: Acquiring associations that MAY or MAY NOT be object labels.

As I noted before: Not only are these dogs gifted, but they are all Border collies – dogs specifically BRED to learn quickly and to understand vocal commands given by humans...and even then, not all their subjects do well. Note that collies are no longer bred specifically for herding, but likely for physical standards of "beauty", so that the genetics no longer hold as strongly, which may be why so few dogs succeed.

I repeat my earlier criticism, that although I DO find the behavior of some of these dogs impressive, I do not agree that their behavior is all that similar to that of children who are learning labels nor that dogs (in general?!--remember only a few Border collies succeed in any kind of label learning at all) can be a model for studying mechanisms involved in human word acquisition. The authors should stick to their truly impressive data (which are impressive!) rather than make additional claims of parallels with human acquisition.

They have backed down a bit and said that more work is needed, but they haven't backed down enough, simply because they do not have the data beyond one dog, Whiskey, to make any such claims – no other dog of theirs has been appropriately tested.

In References: Why are some titles of journal articles capitalized whereas other are not?

Reviewer: 2

Comments to the Author(s)

In this paper, the authors recruited 6 dogs who had previously demonstrated the ability to associate toys with specific names. They then tested their ability to learn new associations of 6 + 12 objects over a short period (1 week), and whether the dogs could retain those associations 1 and 2 months later without any practice in the meantime. I thought the premise was interesting, the design was well-executed, and the conclusions were warranted. I did still have a few questions about the specifics (e.g. subject criteria, methodology) which are addressed below in my line-by-line feedback.

Title: I wonder if would be better to say 'acquisition' as opposed to 'learning rate' since your question is broader and categorical (i.e. yes/no did the dogs learn the words in a set amount of

time) vs. the specifics that 'learning rate' might call to mind (i.e. how many exposures are needed/learning curves/etc.). Also, rather than 'selected' dogs, maybe 'word-knowledgeable' dogs (or some descriptor that defines how they were selected).

Line 15. Might be useful to clarify here that they are retaining in their memory 'without further practice post-acquisition' or something like that?

Lines 59-64. This sentence is hard to parse and parts are redundant. I think it would be easier to understand if rephrased slightly and broken up into 2 sentences. Maybe something like: "In infants, the ability to rapidly learn new words is thought to mark a shift in their thinking, whereby they now realize that words represent objects [15]. Thus, evidence for comparable rapid learning of object names in dogs, combined with previous evidence showing that dogs with vocabulary knowledge relate labels to categories of items, instead of only associating one word to one object [16], may support the idea that dogs, similar to human children, can learn that words represent items [12]."

Subject criteria - how was their knowledge of over 26 object names verified? E.g. owner report or actually tested by an independent party? Were there nominated dogs who were deemed to not actually meet the criteria? I think it would be helpful to either explain in more detail and/or provide a link to the website (geniusdogchallenge.com), which I found by googling the Genius Dog Challenge social media campaign that you first mention in the experimental setup section directly below. The 'How to Apply' section of the website (assuming that is the criteria the 6 current dogs were held to?) was useful in ascertaining how dogs were deemed to meet the criteria.

Line 107-111: obviously sample size is such that we can't draw any firm conclusions, but would be interesting to know which subjects specifically were trained for an avg of 1.5 h and 2.5-5.5 h, respectively

Lines 144-145: why did the owner remove the requested toy if the dog brought the incorrect one? To make it the same # and ratio of test/control items between dogs? Please explain rationale. And what happened with the incorrectly retrieved toy? Was it replaced, or also removed?

Lines 173-176: what was the rationale for replacing the toys after 3 and 9 trials (as opposed to after every trial)? Why did the # of trials between replacement differ between experiments?

Line 190: It would be helpful to present results of experiment 3 and experiment 4 as you do the first 2 experiments... i.e. when you say XX dogs retrieved all toys, remind the reader of the total number of all toys. So, "In Experiment 3, five dogs successfully retrieved all 6 toys" and "In Experiment 4, three dogs successfully retrieved all 6 toys"

Line 191: how many toys did the dog who didn't perform above chance successfully retrieve?

Line 253: ah great - here is the link to the website! Although this also clarifies that the website was used as a recruitment tool AFTER the recruitment of the 6 dogs in the current study (presumably for future studies)... so I think still useful to add more detail in the methods re: recruitment, at the detail level of the 'how to apply' page of the website

Line 266: consider rephrasing to "The similarities in word learning rate (or acquisition??) and retention capacity between infants and 'word-knowledgeable' dogs further suggests that..."

===PREPARING YOUR MANUSCRIPT===

===PREPARING YOUR REVISION IN SCHOLARONE===

-- Ensure that your data access statement meets the requirements at <https://royalsociety.org/journals/authors/author-guidelines/#data>. You should ensure that you cite the dataset in your reference list. If you have deposited data etc in the Dryad repository, please only include the 'For publication' link at this stage. You should remove the 'For review' link.

-- If you have uploaded ESM files, please ensure you follow the guidance at <https://royalsociety.org/journals/authors/author-guidelines/#supplementary-material> to include a suitable title and informative caption. An example of appropriate titling and captioning may be found at [https://figshare.com/articles/Table_S2_from_Is_there_a_trade-off_between_peak_performance_and_performance_breadth_across_temperatures_for_aerobic_sc](https://figshare.com/articles/Table_S2_from_Is_there_a_trade-off_between_peak_performance_and_performance_breadth_across_temperatures_for_aerobic_scope_in_teleost_fishes_/3843624) ope_in_teleost_fishes_/3843624.

Author's Response to Decision Letter for (RSOS-210976.R0)

See Appendix A.

Decision letter (RSOS-210976.R1)

Dear Ms Dror,

It is a pleasure to accept your manuscript entitled "Acquisition and long-term memory of object names in a sample of Gifted Word Learner dogs" in its current form for publication in Royal Society Open Science. The comments of the reviewer(s) who reviewed your manuscript are included at the foot of this letter.

on behalf of Dr Rosalind Arden (Associate Editor) and Kevin Padian (Subject Editor)
openscience@royalsociety.org

Associate Editor Comments to Author (Dr Rosalind Arden):

Comments to the Author:

Thank you very much for working on this manuscript and for incorporating Reviewers comments. You have conducted an interesting study that will no doubt foster further work of this kind in other labs. You deserve congratulations for working around the restrictions caused the pandemic, and for acquiring data in spite of it! Your revision includes good faith engagement with the many helpful Reviewer suggestions and will now make a terrific inclusion in RSOS. Thank you for this submission.

Appendix A

Dear editor and reviewers,

We thank you for the time and thought you have invested in our manuscript. We have addressed your comments and believe that they have helped to significantly improve the manuscript.

Below you can find a point-by-point answer to your comments.

Sincerely yours,

Shany Dror,
Ádám Miklósi,
Andrea Somnese,
Andrea Temesi,
Claudia Fugazza

Comments by: Associate Editor (Dr. Rosalind Arden):

This is a fascinating piece of work and will be of much interest. The two reviewers would like to see some revisions made and so would we.

The work as dog qua dog is good enough to stand alone without making claims that the dog is a good model for learning about the development of language in children. This is nonsensical - there are a super abundance of children from whom we can gather phenotypic and genetic data that directly bear on human language development - we don't need dogs for that - if you have a particular idea in mind for why the dog is a good model or what we could learn from dogs that we cannot learn from human children and their adult relatives, then please say it.

Authors answer: We thank the editor for this comment. We would like to clarify that we do not suggest dogs as a model for developmental aspects (see lines 256-260) but rather for evolutionary, cognitive, and comparative aspects of cognition that are related to language (see the discussion about domain-general learning mechanisms described in lines 267-270). The dog has already been recognized as a model for social cognition, examined in behavioral (Topál et al., 2009) and neuroimaging (Bunford et al., 2017) studies. Here we suggest that GWL dogs expand this possibility to aspects of cognition thought to be related to language in humans.

What is striking and interesting here is the evidence of variation among individuals and of capacity. That this work was done under the restrictions of the pandemic is especially praiseworthy.

I would consider cutting the references to cross-species comparisons, but at the least, please do listen to Reviewer 1 who proposes that the ms still in places goes beyond its evidence. It doesn't need to - the work is great as it is. Reviewer 2 has asked some key questions. Answering those will strengthen the ms.

We hope you will revise this after a careful reading of the Reviewer comments. It will be a splendid paper.

Authors answer: We thank the editor for the kind encouragement. We have addressed the comments of Reviewer 1 (see below the reply to L. 222ff). We rephrased our references to emphasize the differences between the species and explain the potential for comparative research (throughout the manuscript and especially on L. 263-266).

Reviewer comments to Author:

Reviewer: 1

Dror and colleagues have revised their intriguing paper on the capacities of dogs to learn human labels for objects and to retain these labels for several months. I still have a few comments on the paper as the authors have not quite addressed all my concerns.

L. 19: I still object to the comparison between the dogs and young children. As I noted before, even if children have a very small vocabulary before their ‘spurt’, the labels are not restricted to one particular object (a “ball” will be used to refer to a number of different balls, and even sometimes the label is over- or under-extended). To argue that dogs are good models for other nonhumans, however, DOES make sense.

Authors answer: thank the reviewer for this comment. The sentence has been deleted and we have adjusted our claim L. 19-21.

L. 34: Here the authors have shifted from early vocabulary acquisition to children well beyond the beginning of the vocabulary spurt...and then shift back to 2 yr olds...the children here are in very different stages and comparisons that are being made are confusing and conflate many issues.

Authors answer: We have rephrased this paragraph (L. 35-43). It now starts with a sentence explaining the complexity of cross-study comparisons and then follows chronological order.

L. 47: Please examine Hirsh-Pasek & Trieman, “Doggerel: Motherese in a new context”, paper written ~1982 or so.

Authors answer: we thank the reviewer for pointing us in the direction of this reference. It has been added (L. 59-50).

L. 53-55: I would again argue that all these dogs learned was a brief association of sound and object; to call this behavior the learning of names is quite a stretch.

Authors answer: the study did not intend to examine learning mechanisms but rather describe the phenomena of rapid acquisition. We agree with the reviewer that we cannot conclude from our results what were the underlying cognitive mechanisms, so we have rephrased the sentence to use a more general language (L. 60).

L. 59-63: This sentence is quite complex and difficult to parse...please edit.

Authors answer: we apologize, it has been deleted as the paragraph was rephrased.

L. 63: I agree that one dog seems to understand something about categories, which suggests that such learning is possible...but the authors cannot make such claims for the other subjects in the other studies and later for all their subjects.

Authors answer: this paragraph has been rephrased (L. 54-56).

L. 110: 5.5 hrs/day is intensive training! Which dog was that? Did that dog do better than some of the others? Would be an important bit of information! And why are descriptions of Experiment 2 placed in the paragraph devoted to Experiment 1?

Authors answer: the training duration of Experiment 1 and 2 now appear in the corresponding sections (L. 108-117 and L. 123-125 respectively). We apologize for the inconsistency. The dog which received 5.5 hours of training per day was Gaia (this is now specified in the text). As shown in table 1, she did not perform better than the other dogs.

L. 118: How did the experimenters guarantee that the toys were stored away and that the owners did not review things with their dogs? Promises to do so are nice, but people often want their dogs to shine, so....

Authors answer: the owners were instructed to store the toys in the original boxes in which they have arrived but as the experimenters could only monitor the tests and were not present in the houses, we could only rely on the owners' word. We would however like to mention that similar constraints are present also in studies examining infants' word retention, as nothing is prohibiting the parents from practicing with their child at home.

L. 128: There's a certain amount of 'exclusion' occurring here—dog knows it has fetched X already, so maybe knows to ignore X at least for a while? Why not put in a mix of fetched and some other old toys, or maybe a brand new, unnamed toy?

Authors answer: Preliminary observations showed that the dogs participating in this experiments were highly motivated to play with toys, especially new toys. Therefore, we did not provide new toys to prevent such over-excitement, which could have affected the results of the tests.

Supplying additional toys (new or old) would not have prevented the dogs from using exclusion to determine the requested object. However, even if dogs are able to perform an extensive exclusion process (as we always placed back at least three toys at the time they would have needed to remember all three toys) this would still not explain their capacity to choose the correct toy out of the 3 that remained.

L. 163: So here the owner showed the dog the correct toy? Wouldn't that act as a refresher trial?

Authors answer: The owner did not show the dog the toy or say the name of the toys but simply placed the toy away, out of the dogs view. (now explained more clearly in L. 182-182, and 185-186).

L. 175ff: To be truly conservative, the chance level should be 1/6 for the first set and 1/12 for

the second, assuming that the dog would ignore its old toys and just concentrate on the new ones...

Authors answer: In one of the cases, where a dog performed a mistake in Experiment 1, the dog retrieved one of his old toys. Therefore, the possibility of a dog retrieving an old toy exists. In this revised version we recalculated our results, using the most conservative chance level possible and this did not substantially change our findings.

For all of the experiments we ignored the presence of old toys and used the lowest number of new toys from which the dog could choose. In Experiments 1, 3, and 4 the number of new (or re-introduced) toys varied between 6-4. We, therefore, calculated the chance level as $\frac{1}{4}$ (not $\frac{1}{6}$ as the reviewer recommended).

In Experiment 2, the number of new toys from which the dog could choose varied between 12-4. Hence, here again we calculated the chance level as $\frac{1}{4}$ (not $\frac{1}{12}$ as the reviewer recommended).

The methods section has now been adjusted (L. 191-195).

L. 209: In reality, only SOME OF the dogs could perform the task... the authors cannot make a broad generalization.

Authors answer: we have rephrased this sentence (L. 224-225).

L. 222ff: Given that the labels are connected to a SINGLE object, rather than a class (e.g., “balls”, which is what Pilley and Reid tried to test) for the dogs in THIS study, the authors cannot argue for anything other than an efficient associative mechanism... That is all that THEY tested. They cannot argue that their dogs would perform as well as others trained and tested in different paradigms. The authors are correct that the POSSIBILITY exists, but their claims for THEIR data are way too strong.

Authors answer: this paragraph has now been rephrased. We start with an explanation about the differences between infants’ comprehension of object names and that of GWL dogs, scanning the limited studies examining categorization and referential understanding in GWL dogs (L. 260-262). We then emphasize that our intention is not to claim for similarities between infants and GWL dogs but rather to highlight the potential for comparative research (L. 263-266).

L. 237: Acquiring associations that MAY or MAY NOT be object labels.

Authors answer: this paragraph has been deleted.

As I noted before: Not only are these dogs gifted, but they are all Border collies—dogs specifically BRED to learn quickly and to understand vocal commands given by humans...and even then, not all their subjects do well. Note that collies are no longer bred specifically for herding, but likely for physical standards of “beauty”, so that the genetics no longer hold as strongly, which may be why so few dogs succeed.

Authors answer: we thank the reviewer for this comment. We have recently published a study (Fugazza et al., 2021b) specifically on the extreme variation in this capacity between typical dogs (including typical Border collies) and dogs with a vocabulary of object names. In the current manuscript, we have now emphasized the differences between typical dogs and dogs with this capacity by using the term Gifted Word Learner (GWL) dogs, when referring to dogs with the vocabulary of object names. Regarding the dogs' breed, in our previous reply, we have referred to studies about GWL dogs that were not Border Collies (BC)(Fugazza et al., 2021a; Griebel and Oller, 2012; Ramos and Ades, 2012). Moreover, from the 15 dogs that were recruited as a result of our social media campaign, 4 are not BC (1 Pekingese, 1 German Shepherd, 1 Miniature Australian Shepherd, 1 mongrel).

We agree with the reviewer that genetics might serve as a partial explanation for the ability of GWL dogs to rapidly learn new object names. However, breed alone cannot account for this difference. In our latest study (Fugazza et al., 2021b) we found that BC were not typically able to learn the names of objects.

I repeat my earlier criticism, that although I DO find the behavior of some of these dogs impressive, I do not agree that their behavior is all that similar to that of children who are learning labels nor that dogs (in general?!--remember only a few Border collies succeed in any kind of label learning at all) can be a model for studying mechanisms involved in human word acquisition. The authors should stick to their truly impressive data (which are impressive!) rather than make additional claims of parallels with human acquisition.

They have backed down a bit and said that more work is needed, but they haven't backed down enough, simply because they do not have the data beyond one dog, Whiskey, to make any such claims—no other dog of theirs has been appropriately tested.

Authors answer: we thank the reviewer for giving us a second opportunity to clarify our claims. As now explained in L. 263-266 and L. 272-274 we do not claim that the cognitive mechanisms are similar but rather that there is much to be learned from comparative research.

In References: Why are some titles of journal articles capitalized whereas other are not?

Authors answer: we apologize for the inconsistency. It has now been corrected.

Reviewer: 2

Comments to the Author(s)

In this paper, the authors recruited 6 dogs who had previously demonstrated the ability to associate toys with specific names. They then tested their ability to learn new associations of 6 + 12 objects over a short period (1 week), and whether the dogs could retain those associations 1 and 2 months later without any practice in the meantime. I thought the premise was interesting, the design was well-executed, and the conclusions were warranted. I did still have a few questions about the specifics (e.g., subject criteria, methodology) which are addressed below in my line-by-line feedback.

Title: I wonder if would be better to say ‘acquisition’ as opposed to ‘learning rate’ since your question is broader and categorical (i.e., yes/no did the dogs learn the words in a set amount of time) vs. the specifics that ‘learning rate’ might call to mind (i.e., how many exposures are needed/learning curves/etc.). Also, rather than ‘selected’ dogs, maybe ‘word-knowledgeable’ dogs (or some descriptor that defines how they were selected).

Authors answer: we thank the reviewer for this suggestion and have changed the title of our manuscript.

Line 15. Might be useful to clarify here that they are retaining in their memory ‘without further practice post-acquisition’ or something like that?

Authors answer: this has been added (L. 16).

Lines 59-64. This sentence is hard to parse and parts are redundant. I think it would be easier to understand if rephrased slightly and broken up into 2 sentences. Maybe something like: “In infants, the ability to rapidly learn new words is thought to mark a shift in their thinking, whereby they now realize that words represent objects [15]. Thus, evidence for comparable rapid learning of object names in dogs, combined with previous evidence showing that dogs with vocabulary knowledge relate labels to categories of items, instead of only associating one word to one object [16], may support the idea that dogs, similar to human children, can learn that words represent items [12].”

Authors answer: The sentence was deleted as the paragraph was rephrased.

Subject criteria – how was their knowledge of over 26 object names verified? E.g., owner report or actually tested by an independent party? Were there nominated dogs who were deemed to not actually meet the criteria? I think it would be helpful to either explain in more detail and/or provide a link to the website (geniusdogchallenge.com), which I found by googling the Genius Dog Challenge social media campaign that you first mention in the experimental setup section directly below. The ‘How to Apply’ section of the website (assuming that is the criteria the 6 current dogs were held to?) was useful in ascertaining how dogs were deemed to meet the criteria.

Authors answer: we have added a reference to a study that examined the same 6 dogs (Fugazza et al., 2021b) and determined their knowledge. A brief summary of the procedure is available on L. 81-86.

Line 107-111: obviously sample size is such that we can’t draw any firm conclusions, but would be interesting to know which subjects specifically were trained for an avg of 1.5 h and 2.5-5.5 h, respectively.

Authors answer: we have added the names of the dogs to the description of the training duration (L. 108-111 and L. 123-125).

Lines 144-145: why did the owner remove the requested toy if the dog brought the incorrect one? To make it the same # and ratio of test/control items between dogs? Please explain rationale. And what happened with the incorrectly retrieved toy? Was it replaced, or also

removed?

Authors answer: Studies on the ability of dogs to retrieve named objects commonly give the dogs a second chance if they have committed a mistake (Kaminski et al., 2009, 2008) and do not include the outcomes of second chances in the data analysis. We observed that dogs tend to remember which toy they have failed to retrieve and would often bring this toy in the next trial, even if the owner requested a different one. Therefore, in experiments 1 and 2, we gave the dog a second chance and did not include it in the data analysis. However, in the memory test, we did not want to supply a second chance, as this could have served as a memory refreshment. We removed the toy to prevent the dog from making a mistake in the next trial. As now explained in the revised text (L. 135-137, 145-147, 159-164), we maintained the chance level by placing the toys back whenever there were only 3 reintroduced toys left on the floor.

Lines 173-176: what was the rationale for replacing the toys after 3 and 9 trials (as opposed to after every trial)? Why did the # of trials between replacement differ between experiments?

Authors answer: We did not place the toy back after every trial because this would have interrupted the experiment very often and required the dog to wait while the owner leaves and goes to the other room. We had observed that dogs tend to become stressed or over-excited in this situation. Therefore, when possible, we preferred not to interrupt the dog's work. In addition, we wanted to have a consistent chance level for all experiments. This meant that the minimum number of toys on the floor was kept at 4 for all experiments (and chance level was conservatively set at 25%, as if the number of toys to choose from was always 4). Consequently, the number of trials between replacements changed according to the number of toys tested in each experiment.

Line 190: It would be helpful to present results of experiment 3 and experiment 4 as you do the first 2 experiments... i.e., when you say XX dogs retrieved all toys, remind the reader of the total number of all toys. So, "In Experiment 3, five dogs successfully retrieved all 6 toys" and "In Experiment 4, three dogs successfully retrieved all 6 toys".

Authors answer: thank you for drawing our attention to this. It has been changed (L. 204, 206).

Line 191: how many toys did the dog who didn't perform above chance successfully retrieve?

Authors answer: in the revised version we have implemented a stricter analysis with a higher chance level. Therefore, now two dogs have not performed above chance. One of these dogs retrieved 3 toys and the other 5 (L. 204-208).

Line 253: ah great - here is the link to the website! Although this also clarifies that the website was used as a recruitment tool AFTER the recruitment of the 6 dogs in the current study (presumably for future studies) ... so I think still useful to add more detail in the methods re: recruitment, at the detail level of the 'how to apply' page of the website.

Authors answer: we have now added the link also to the methods section (L. 94).

Line 266: consider rephrasing to "The similarities in word learning rate (or acquisition??) and retention capacity between infants and 'word-knowledgeable' dogs further suggests that..."

Authors answer: this part of the sentence has been deleted as a consequence of rewriting the paragraph.

References

- Bunford, N., Andics, A., Kis, A., Miklósi, Á., Gácsi, M., 2017. *Canis familiaris* as a model for non-invasive comparative neuroscience. *Trends Neurosci.* 40, 438–452. <https://doi.org/10.1016/j.tins.2017.05.003>
- Fugazza, C., Andics, A., Magyar, L., Dror, S., Zempléni, A., Miklósi, Á., 2021a. Rapid learning of object names in dogs. *Sci. Rep.* 11, 1–11. <https://doi.org/10.1038/s41598-021-81699-2>
- Fugazza, C., Dror, S., Sommese, A., Temesi, A., Miklósi, Á., 2021b. Word learning dogs (*Canis familiaris*) provide an animal model for studying exceptional performance. *Sci. Rep.* 11, 1–9. <https://doi.org/10.1038/s41598-021-93581-2>
- Griebel, U., Oller, D.K., 2012. Vocabulary learning in a Yorkshire terrier: Slow mapping of spoken words. *PLoS One* 7, 1–10. <https://doi.org/10.1371/journal.pone.0030182>
- Kaminski, J., Fischer, J., Call, J., 2008. Prospective object search in dogs : mixed evidence for knowledge of What and Where. *Anim. Cogn.* 11, 367–371. <https://doi.org/10.1007/s10071-007-0124-1>
- Kaminski, J., Tempelmann, S., Call, J., Tomasello, M., 2009. Domestic dogs comprehend human communication with iconic signs. *Dev. Sci.* 12, 831–837. <https://doi.org/10.1111/j.1467-7687.2009.00815.x>
- Ramos, D., Ades, C., 2012. Two-Item Sentence Comprehension by a Dog (*Canis familiaris*). *PLoS One* 7, 1–7. <https://doi.org/10.1371/journal.pone.0029689>
- Topál, J., Miklósi, Á., Gácsi, M., Dóka, A., Pongrácz, P., Kubinyi, E., Virányi, Z., Csányi, V., 2009. The Dog as a Model for Understanding Human Social Behavior. *Adv. Study Behav.* 39, 71–116. [https://doi.org/10.1016/S0065-3454\(09\)39003-8](https://doi.org/10.1016/S0065-3454(09)39003-8)